# Mapping Functional Language Areas with non-Functional Brain MRI

**Omri Leshem**[1]                    OMRILESHEM@MAIL.TAU.AC.IL
**Atira Sara Bick**[2]                    BICA@HADASSAH.ORG.IL
**Nahum Kiryati**[3]                    NAHUM.KIRYATI@GMAIL.COM
**Netta Levin**[2]                    NETTA@HADASSAH.ORG.IL
**Arnaldo Mayer**[4]                    ARNMAYER@GMAIL.COM

[1] *School of Electrical and Computer Engineering, Tel-Aviv University, Israel*

[2] *fMRI Unit, Department of Neurology, Hadassah Medical Center and Faculty of Medicine, Hebrew University of Jerusalem, Israel*

[3] *The Manuel and Raquel Klachky Chair of Image Processing, School of Electrical and Computer Engineering, Tel-Aviv University, Israel*

[4] *Diagnostic Imaging, Sheba Medical Center, affiliated to the School of Medicine, Tel Aviv University, Israel*

**Editors:** Accepted for publication at MIDL 2025

## Abstract

Mapping eloquent brain areas has become a standard of care in brain surgery. Current imaging-based techniques usually rely on functional MRI (fMRI), which measures neural activity via the blood oxygenation level-dependent signal. fMRI protocols are time-intensive, require active patient collaboration, and involve laborious manual post-processing and expertise, making them difficult to implement in some clinical scenarios. In this research, we propose a fully automated deep neural pipeline for the mapping of Broca and Wernicke functional language areas using multiple non-functional MRI modalities. The proposed method is evaluated on a cohort of 30 drug-resistant epilepsy patients, showing encouraging qualitative and quantitative results and suggesting its potential applicability as an effective and practical tool for neurosurgical planning and navigation. Implementation details can be found in our GitHub.

**Keywords:** Deep Learning, Multi-modal, fMRI, Brain Mapping, Neurosurgical Planning

## 1. Introduction

The mapping of functional language areas, specifically Broca and Wernicke, is a crucial step in neurosurgical procedures to ensure patient safety. These regions are integral to language processing and communication, with Broca primarily responsible for word generation, and Wernicke essential for word comprehension. Accurate localization of these areas is critical for preoperative planning and intraoperative navigation, especially in surgeries involving the temporal and frontal lobes (Sanai et al., 2008).

Harming the Broca and Wernicke regions can lead to serious disabilities, such as aphasia or impaired language comprehension (Collee et al., 2022), and highlights the need to achieve high sensitivity with localization methods.

Functional MRI (fMRI) remains the imaging gold standard for the identification of functional language areas, measuring neural activity through task-based blood oxygen level-dependent (BOLD) signal variations (Price, 2000). However, fMRI is associated with significant practical challenges. It usually requires a level of patient collaboration that may not be possible for pediatric and elderly populations, or patients specifically impaired by the targeted lesion. For example, a lesion located near Broca may affect the patient's ability to perform a cognitive task required to activate the very same Broca in an fMRI mapping paradigm.

Moreover, fMRI is prone to noise, motion artifacts, and tumor-induced signal perturbations, reducing its reliability and accuracy in some clinical scenarios (Constable, 2023) (Liu, 2016) (Schleim and Roiser, 2009). In practice, safety margins are often added around fMRI localizations to improve patient protection during surgery (Riley et al., 2022) (Wang et al., 2019), with a 5 mm margin often considered a reasonable choice.

To address fMRI limitations, complementary approaches based on non-functional MRI modalities, such as T1-weighted (T1w) imaging and diffusion tensor imaging (DTI) are becoming of interest (Ekstrand et al., 2020) (Son et al., 2019) (Ellis and Aizenberg, 2022). These modalities are acquired in standard clinical MRI workflows and are more accessible than fMRI protocols. Specifically, DTI provides precious white matter connectivity information. Notably, the DTI reconstruction of the arcuate fasciculus that connects between Broca and Wernicke, may prove useful in their spatial localization. (Son et al., 2019)

While initial studies have demonstrated the feasibility of structural MRI-based approaches, their performance has been limited. For example, in (Ekstrand et al., 2020), linear regression was used to predict whole-brain functional activation during reading tasks from structural imaging data. However, the linear model imposed limitations, and functional area segmentation was not addressed.

In another study (Son et al., 2019), a convolutional neural network (CNN) was used to synthesize diffusion tensor (DT) data from fMRI scans, demonstrating inter-modality correlation. However, this approach focused on generating a simpler modality (DT) from a more complex one (fMRI) and did not attempt direct localization of language regions, limiting its applicability.

Similarly, in (Ellis and Aizenberg, 2022), a U-Net-like architecture (Ronneberger et al., 2015) was used to predict fMRI activation patterns from T1w scans across 42 cognitive tasks using the HCP dataset (Van Essen et al., 2012). (Ellis and Aizenberg, 2022) have also shown that incorporating T1w with diffusion tensor (DT) data improved the results. While this method outperformed atlas-based approaches, it lacked extensive quantitative validation in clinical settings, and brain mapping was not performed.

The findings of (Ekstrand et al., 2020) (Son et al., 2019) (Ellis and Aizenberg, 2022) highlight the potential of structural imaging and inter-modality prediction for functional area mapping while underscoring the need for more advanced methods that integrate anatomical and connectivity data for accurate localization. Furthermore, validation on clinical datasets is essential to verify the approach's effectiveness in real-world clinical applications and settings.

In this research, we propose a novel deep neural framework for the localization of Broca and Wernicke areas that combines detailed anatomical information, provided by T1w scans,

with spatial connectivity information, provided by DTI diffusion directionality map.

**The main contributions of this paper are as follows:**

1. Introducing a multi-modal deep neural pipeline for localizing the Broca and Wernicke areas using non-fMRI input scans. The method is focused on sensitivity, to minimize the risk of missing critical areas.

2. Extensive validation and assessment on 30 real clinical cases: drug resistant epilepsy patients.

3. Detailed uncertainty quantification and safety margin analysis.

## 2. Methods

A block diagram of the proposed method is shown in Figure 1. We adopt the approach of (Ellis and Aizenberg, 2022), (Nelkenbaum et al., 2020) and design a framework that jointly processes T1w and DWI data for segmenting the Broca and Wernicke functional areas. The input to our neural network consists of two MRI scans: a T1w scan, which provides high-resolution anatomical description and strong contrast between brain tissues, and a Diffusion Encoded Colored (DEC) map (Pajevic and Pierpaoli, 1999), derived from DTI, encoding white matter tracts connectivity. The desired output of the network is a segmentation map of the Broca and Wernicke areas. The ground truth is manually extracted from task-based fMRI language activation maps, ensuring the accurate labeling of functional language areas.

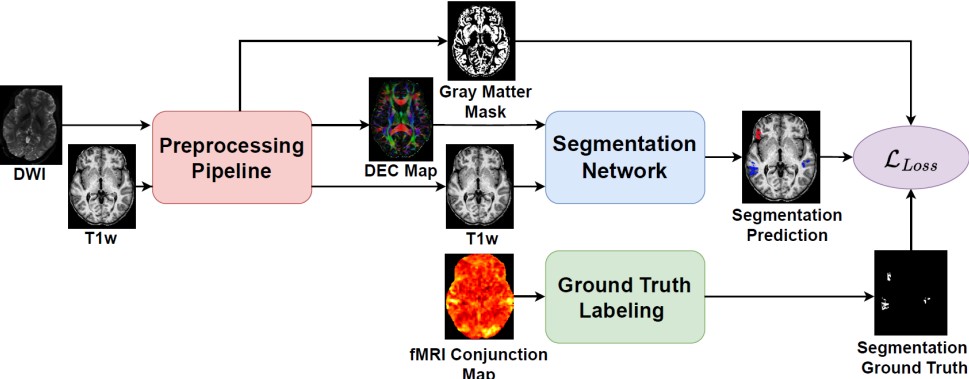

Figure 1: Block diagram of the proposed framework, outlining the workflow from data acquisition and labeling to training and final segmentation. Further details are provided in Sections 2.1, 2.1.1, 2.3

### 2.1. Pre-processing

A 9 parameters similarity registration (rotation, translation and anisotropic scaling) was performed between corresponding DWI and T1w scans using the FSL toolkit (Jenkinson and Smith, 2001) (Jenkinson et al., 2002). Brain extraction was performed with

ANTsPyNet (Cullen and Avants, 2018). Gray matter (GM) was segmented from T1w scans also with FSL (Zhang et al., 2001). DTI fitting and DEC map generation were carried out using the DIPY package (Garyfallidis et al., 2014).

### 2.1.1. Ground Truth Labeling Process

Analysis of the raw functional data was performed in BrainVoyager (Goebel et al., 2006). Head motion correction, slice scan correction and high-pass temporal filtering in the frequency domain were applied in order to remove drifts and to improve the signal-to-noise ratio. Functional scans were aligned to the anatomical (T1w) scans, after which both anatomical and functional scans were realigned to ACPC space. A general linear model (GLM) was used to create a conjunction t-map for three classical language tasks: Visual verb generation, Auditory verb generation, and Sentence generation. Binary functional ground-truth maps were consecutively derived by applying, for each patient, a pragmatic individual threshold to the language conjunction t-map. The latter was manually tuned using ITK-SNAP (Yushkevich et al., 2006) to retain activations solely within known language-related regions, focusing on GM patterns to ensure alignment with the anatomical boundaries of the Broca and Wernicke regions. The entire process was conducted under the close supervision and verification of an fMRI expert, following the same methodology applied at our institution for the generation of neuro-surgical navigation maps.

## 2.2. Segmentation Network Architectures

Two network architectures were considered for the segmentation task: 1) AGYnet (Nelkenbaum et al., 2020): a convolutional network specifically designed for multimodal segmentation. It features two separate encoders, one for each modality, and a joint decoder utilizing attention gating (AG) mechanisms. 2) Swin-UNETR (Tang et al., 2022): a Swin transformer encoding path followed by a fully convolutional decoding path. The encoding-decoding of the Swin-UNETR architecture utilized the down-scaling/up-scaling and skip connections of a U-Net architecture. In the Swin-UNETR architecture, input modalities were concatenated along the channel axis at the network's entrance. The Swin-UNETR architecture was trained in a multiclass setting with three output channels (background, Broca, Wernicke). The AGYNet architecture was trained in a single-class mode with two output channels (background and Broca or Wernicke), where separate models were trained independently for Broca and Wernicke. In both architectures, a softmax activation function was applied at the final layer to generate probability-like outputs, representing the likelihood of each voxel belonging to a specific class. The argmax operator was subsequently employed as a decision rule to convert probability values into binary segmentations. Training AGYNet in a multiclass setting was unstable and failed to converge.

## 2.3. Training

Training was implemented in two steps: The network was initially pre-trained on the HCP dataset (Van Essen et al., 2012) for a related segmentation task, specifically the segmentation of Brodmann areas 44-45 and 22. These areas strongly correlate with the anatomical definitions of the Broca and Wernicke regions and are highly associated with functional language areas. The anatomical cortical labeling of Brodmann areas was automatically

generated using the FreeSurfer pipeline (Fischl, 2012). The classical Dice loss (Huang et al., 2018) was used for this step.

In the second step, the networks were fine-tuned on a smaller target dataset based on fMRI activations ground-truth. Left-right brain flip augmentations were randomly applied during training to address the over-representation existing in the general population for left language dominance. We also introduced an Anatomically-Guided Loss function for the fine-tuning step that is detailed in the following subsection.

## 2.4. Anatomically-Guided Loss Function

The training loss is defined as the sum of two (equally-weighted) components. The first one, enforcing overlap between spatial segmentation and ground-truth, is the well-known DiceCE loss (Taghanaki et al., 2019), denoted as $\mathcal{L}_{\text{DiceCE}}$.
The second component penalizes predictions falling outside the gray matter (GM), leveraging the prior knowledge that all functional activations are, by definition, within the GM. Accordingly, we define the anatomical loss, $\mathcal{L}_{\text{Anatomy}}$ as,

$$\mathcal{L}_{Anatomy} = 1 - \frac{\sum_i \mathrm{p}_i^2 \cdot \mathrm{gm}_i^2}{\sum_i \mathrm{p}_i^2} \tag{1}$$

where $p_i$ denotes a voxel from the predicted segmentation, taking probability values between 0 and 1, and $gm_i$ represents the corresponding voxel from the GM segmentation mask, which has binary values. The summation is conducted over all the voxels in the image. This term penalizes predictions that extend outside the GM by comparing the squared values of predictions within the GM to the squared values of predictions across the entire brain.

The combined loss function is as follows:

$$\mathcal{L}_{loss} = \mathcal{L}_{DiceCE} + \mathcal{L}_{Anatomy} \tag{2}$$

The impact and justification of $\mathcal{L}_{Anatomy}$ are further detailed in Appendix A.

## 2.5. Construction of the Broca and Wernicke Atlases

As part of a comparative experiment, the proposed method is compared to an atlas-based segmentation scheme. For this purpose, we used the SENSAAS atlas (Labache et al., 2019), which defines cortical areas based on fMRI data. The SENSAAS Atlas was created from scans of 144 healthy right-handed individuals, making it well-suited for our study. The SENSAAS Atlas provides a parcellation of the language areas (Appendix B). For our comparison, we define the Broca area as the union of F301, F3t, INSa3, and f2_2, while the Wernicke area is considered the union of STS3, STS4, STS2, T2_4, SMG7, and AG2. These unions closely match the labeling in our dataset.

## 3. Experiments

### 3.1. Dataset

The dataset used for the experiments consists of 30 clinical cases. In each case, the scans were performed as part of the preoperative evaluation of epilepsy patients at the fMRI unit of our institution. All patients suffered from drug-resistant epilepsy and were referred for an fMRI-DTI protocol to identify language lateralization, fiber location and other functional relevant data as part of the preparation for potential surgery. Patients with low-quality functional scans due to motion artifacts or limited cooperation were not included. Almost all patients did not have clear brain lesions. For our experiment, we utilized the diffusion-weighted imaging (DWI), T1-weighted (T1w), and fMRI scans. The study was authorized by the IRB at our institution.

#### 3.1.1. Data Acquisition

The scans were acquired using a SIGNA Premier GE 3T MRI with a 48-channel coil. T1w scans used an MPRAGE sequence (TR: 2364 ms; TE: 2.95 ms; voxel size: 1mm). DWI scans employed a diffusion sequence (TR: 4500 ms; TE: 56.4 ms; slice thickness: 2mm; resolution: 1x1 mm; b-value: 1000; 64 directions). Functional maps were created by combining activations from Visual/Auditory verb generation and Sentence generation tasks. fMRI data used an EPI sequence (Verb generation: TR: 3000 ms; TE: 30 ms; slice thickness: 2.5mm; resolution: 1.88x1.88; 52 slices; Sentence generation: TR: 2000 ms; TE: 30 ms; slice thickness: 2.5mm). All tasks were performed in the patients' native language.

### 3.2. Training

The two-step training approach described in Section 2.3 was implemented using the HCP dataset (Van Essen et al., 2012), consisting of 694 cases with T1w and DEC scans and binary segmentation maps for Brodmann areas. Segmentation maps for areas 44-45 (Broca) and 22 (Wernicke) served as ground truth. Pre-training used 100 epochs, a batch size of 1, and a constant learning rate of $1 \times 10^{-5}$. Fine-tuning on 30 fMRI epilepsy cases (Section 3.1) segmented *functional* Broca and Wernicke areas using 70 epochs, a batch size of 1, and a cyclic learning rate (cyclicLR) with an average of $5 \times 10^{-5}$. Both steps used the Adam optimizer (Kingma and Ba, 2014), PyTorch (Paszke et al., 2019), and MONAI (Cardoso et al., 2022) frameworks on an NVIDIA RTX 3090 GPU (24 GB). A 5-fold cross-validation scheme was applied with a train (21)-validation (3)-test (6) split, ensuring all cases were tested across folds.

### 3.3. Comparison to Atlas-Based Methods

For comparison with the atlas-based segmentation, we use the atlas described in Subsection 2.5. A fully parametrized 3D affine transformation is applied using the *SimpleITK* toolbox (Beare et al., 2018), transforming the atlas to each patient's coordinate system. Since the atlas contains activation of the left hemisphere only, we limit performance evaluation to the left hemisphere. The latter is identified using brain masks generated using ANTsPyNet (Cullen and Avants, 2018), leveraging data alignment to the ACPC coordinate

system. To reduce the variance caused by minor registration discrepancies, we repeat the process three times and average the results.

## 4. Results

We assessed the performance of the proposed method from both a classification and a segmentation perspective. Figure 2 presents the "Sensitivity vs. 1-Specificity" Receiver Operating Characteristic (ROC) curves, with areas under the curve (AUC) of 0.95/0.96 (Swin-UNETR/AGYnet) for Broca and 0.88 for both architectures for Wernicke. The curves were generated using scikit-learn Python package and restricted to brain voxels identified with a mask generated by ANTsPyNet (Cullen and Avants, 2018) for accuracy. To analyze segmentation performance, segmentation maps were generated by applying an argmax operator between the Broca/Wernicke/Background output channels. The resulting operating point (Swin-UNETR architecture) corresponds to a sensitivity/specificity of 0.58/0.99 for Broca and 0.43/0.99 for Wernicke. The average Dice coefficient was then computed. The quantitative results are summarized in Table 1. The comparison results with the atlas-based method (Section 3.3) are presented in Table 2. Qualitative examples of the predicted segmentation for this operating point are shown in Figure 3.

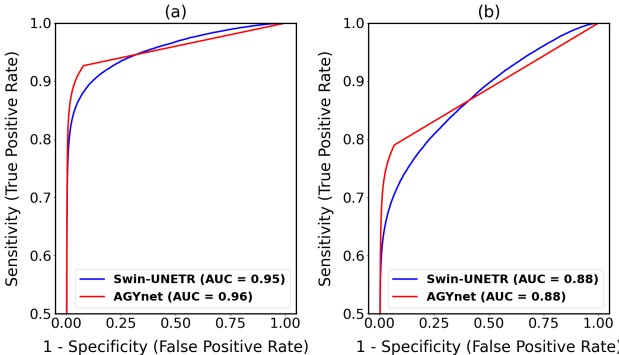

Figure 2: ROC curves (1-specificity vs. sensitivity) of: (a) Broca, (b) Wernicke.

| Model | Broca | | Wernicke | |
|---|---|---|---|---|
| | **AUC** | **Dice Coefficient** | **AUC** | **Dice Coefficient** |
| **AGYnet** | 0.96 | $0.35 \pm 0.03$ | 0.88 | $0.33 \pm 0.04$ |
| **Swin-UNETR** | 0.95 | $0.37 \pm 0.04$ | 0.88 | $0.35 \pm 0.04$ |

Table 1: Comparison of Swin-UNETR and AGYnet models with Dice Coefficient, Sensitivity, and AUC metrics for Broca and Wernicke segmentation.

Although the predicted Dice coefficient is relatively low, it is evident that all predicted activations are confined to compact, characteristic anatomical patterns that align with the anatomical definitions of the Broca and Wernicke areas. The comparison with the atlas-based method reveals the clear superiority of the proposed approach which provides subject-specific predictions. Furthermore, the missing predictions for the proposed method

| AGYnet | | Swin-UNETR | | Atlas | |
|---|---|---|---|---|---|
| **Broca** | **Wernicke** | **Broca** | **Wernicke** | **Broca** | **Wernicke** |
| $0.30 \pm 0.06$ | $0.29 \pm 0.04$ | $0.29 \pm 0.07$ | $0.29 \pm 0.06$ | $0.09 \pm 0.04$ | $0.09 \pm 0.06$ |

Table 2: Dice Coefficient measures only on the left hemisphere for AGYnet, Swin-UNETR, and Atlas-based segmentation.

are situated in close proximity to the actual predictions, as shown in Figure 3, suggesting potential for further refinement and improvement. In order to further support this analysis, segmentation uncertainty and safety margins are investigated and quantified in the next subsection.

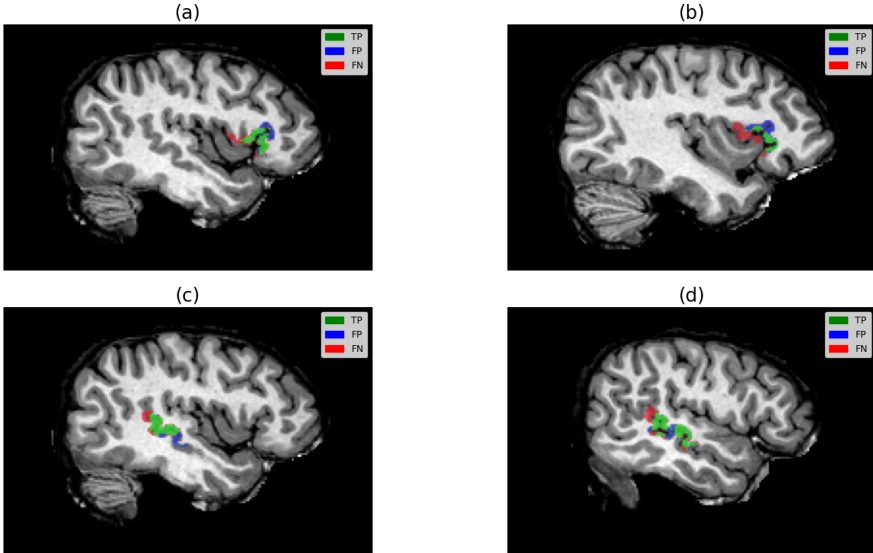

Figure 3: Qualitative Segmentation Sample Slices for the Broca and Wernicke Areas (Swin-UNETR architecture): (a) & (b) Broca at different slices, (c) & (d) Wernicke at different slices. TP, FP, and FN correspond to true positives, false positives, and false negatives, respectively.

### 4.1. Uncertainty Quantification & Sensitivity Margin Analysis

The addition of safety margins is often used in neurosurgical planning and navigation to address segmentation errors and enhance sensitivity (Riley et al., 2022) (Wang et al., 2019). In this context, we applied margin analysis to the best performing algorithm (Swin-UNETR-based) to gain a deeper understanding of its performances. For each prediction, sensitivity was measured as a function of the safety margin size. This is a reasonable assessment because the clinical impact of segmentation errors often depends on accurately capturing critical regions within an acceptable margin, ensuring essential functional areas are not overlooked during planning and surgery.

To perform safety margin analysis, we computed the distance transform from the predictions positions (argmax operating point) and thresholded the resulting map using increasing margin sizes. Sensitivity was then measured for each margin size to evaluate the algorithm's performance. Given the $1\,\text{mm}^3$ spatial resolution of our scans, a margin of one voxel corresponds to a $1\,\text{mm}$ safety margin. We considered margin sizes up to $5\,\text{mm}$, consistent with values common in neurosurgical procedures (Wang et al., 2019).

## Sensitivity

| Margin | Broca | Wernicke |
|--------|-------|----------|
| 1 Voxel | 0.74 | 0.56 |
| 2 Voxels | 0.80 | 0.63 |
| 3 Voxels | 0.87 | 0.71 |
| 4 Voxels | 0.91 | 0.76 |
| 5 Voxels | 0.95 | 0.80 |

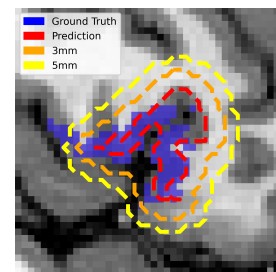 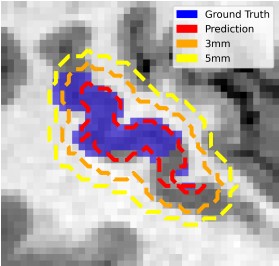

Figure 4: Sensitivity Analysis for the Broca & Wernicke Areas at Varying Margin Widths

We observe (Figure 4, table) that adding a safety margin of two voxels ($= 2\,\text{mm}$) around the predictions brings sensitivity to 0.8 and 0.63 for Broca and Wernicke, respectively. Further increasing the margin to 5 voxels ($= 5\,\text{mm}$), improves even more, with sensitivity reaching 0.95 and 0.80 for Broca and Wernicke, respectively. The effect of the safety margin is shown in Figure 4 for a sample Broca (left) and Wernicke (right) area: the fMRI-based ground truth (blue) is overlaid with the prediction contour (red) and its extensions for a $3\,\text{mm}$ (orange) and a $5\,\text{mm}$ (yellow) safety margin width. We observe that in both examples, for a safety margin $\leq 5\,\text{mm}$, almost all the ground truth pixel are contained within the prediction contour augmented by the safety margin.

## 5. Conclusions & Discussion

We proposed a fully automated deep learning framework to localize the functional Broca and Wernicke regions from multi-modal, non-functional MRI. The model was validated against fMRI-based ground truth from real clinical data, achieving AUCs of 0.96 (Broca) and 0.88 (Wernicke). With a 5mm safety margin—relevant for surgical planning—sensitivities reached 0.95 and 0.80, respectively. While Dice scores remain modest, spatial predictions aligned with anatomical expectations.

To our knowledge, no prior work has addressed the task of mapping functional areas without relying on fMRI, and no clinical dataset or benchmark exists—positioning this study as a pioneering contribution. The core challenge lies in segmenting "invisible" functional regions not directly observable in T1w or DWI, requiring models to learn cross-modality patterns that encode functional activation. This work lays a solid foundation for future research in functional neuroanatomy using non-functional imaging.

## Acknowledgments

We would like to express our sincere gratitude to Mr. Romario Zarik from the School of Electrical and Computer Engineering at Tel Aviv University, Israel, for his valuable assistance in creating a GitHub repository for our research.

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

## Appendix A. Impact of the Anatomically-Guided Loss Function

The main intuition behind the $\mathcal{L}_{Anatomy}$ loss, as described in 2.4, is that differentiating between brain tissues may enhance segmentation accuracy, particularly in regions with complex and intricate anatomical patterns.

To assess the impact of incorporating the $\mathcal{L}_{Anatomy}$ loss, we conducted the experiment outlined in 2.3 twice: first using the $\mathcal{L}_{DiceCE}$ loss (Taghanaki et al., 2019), and then with the addition of the $\mathcal{L}_{Anatomy}$ loss component. Both experiments followed the two-stage training protocol and were conducted using both AGYnet (Nelkenbaum et al., 2020) and Swin-UNETR (Tang et al., 2022). The 5-fold cross-validation results for the Broca and Wernicke regions are reported in Table 3.

The results indicate that the addition of $\mathcal{L}_{Anatomy}$ loss enhances performance across both architectures, particularly for the Broca region, with notable increases in sensitivity and improvements in the Dice coefficient. Although the results for the Wernicke region are

|  | AGYnet | | Swin-UNETR | |
|---|---|---|---|---|
|  | **Without** $\mathcal{L}_{Anatomy}$ | **With** $\mathcal{L}_{Anatomy}$ | **Without** $\mathcal{L}_{Anatomy}$ | **With** $\mathcal{L}_{Anatomy}$ |
| **Broca** | | | | |
| **Sensitivity** | $0.46 \pm 0.09$ | $\mathbf{0.48 \pm 0.08}$ | $0.56 \pm 0.10$ | $\mathbf{0.58 \pm 0.08}$ |
| **Specificity** | $0.99 \pm 0.00$ | $0.99 \pm 0.00$ | $0.99 \pm 0.00$ | $0.99 \pm 0.00$ |
| **Dice Coefficient** | $0.34 \pm 0.03$ | $\mathbf{0.35 \pm 0.03}$ | $0.35 \pm 0.04$ | $\mathbf{0.37 \pm 0.04}$ |
| **Wernicke** | | | | |
| **Sensitivity** | $\mathbf{0.41 \pm 0.07}$ | $0.40 \pm 0.07$ | $\mathbf{0.54 \pm 0.05}$ | $0.43 \pm 0.07$ |
| **Specificity** | $0.99 \pm 0.00$ | $0.99 \pm 0.00$ | $0.99 \pm 0.00$ | $0.99 \pm 0.00$ |
| **Dice Coefficient** | $0.32 \pm 0.04$ | $\mathbf{0.33 \pm 0.02}$ | $0.35 \pm 0.04$ | $0.35 \pm 0.04$ |

Table 3: Impact of the loss function on Dice Coefficient for AGYnet and Swin-UNETR.

mixed, with a decrease in sensitivity, they still show an overall improvement in the Dice coefficient.

We believe that the complex anatomical patterns associated with the Broca region, compared to the Wernicke region, make it more amenable to anatomical guidance, as reflected in the numerical results. These findings further support the potential of $\mathcal{L}_{Anatomy}$ in reinforcing its role as a crucial component for segmentation accuracy.

## Appendix B. SENSAAS Atlas and ROI Definitions

Figure 5, adapted from (Labache et al., 2019, p. 865), illustrates the regions of interest (ROIs) in the original atlas.

We define the Broca area as the union of F3O1, F3t, INSa3, and f2_2, while the Wernicke area is considered the union of STS3, STS4, STS2, T2_4, SMG7, and AG2, as suggested by a senior neuroscientist (A.S.B). These unions closely align with the labeling in our dataset.

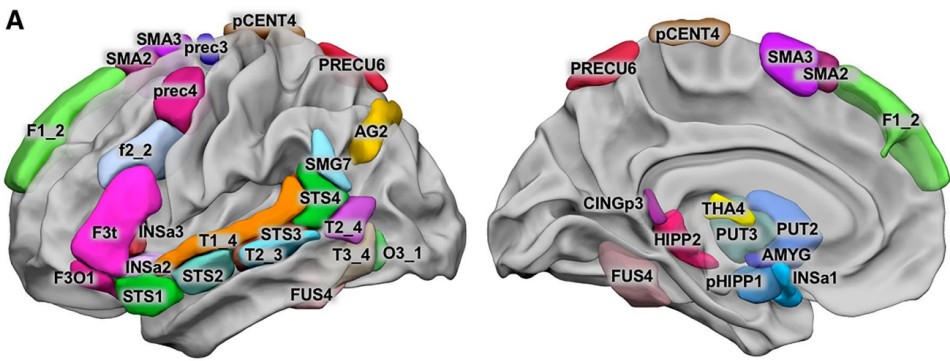

Figure 5: Visualization of the left lateral view of 3D surface renderings for the 32 SENSAAS atlas ROIs, adapted from (Labache et al., 2019, p. 865).

