# OpenReview forum: "Mapping Functional Language Areas with non-Functional Brain MRI"
_MIDL.io/2025/Conference — MIDL 2025 Poster_

### Official Review · Reviewer_Xk6b · 2025-02-15

**Confidence:** 5
**Preliminary Rating:** 4
**Recommendation:** Poster
**Final Rating:** 4

**Summary:**

This paper introduces a method for localizing two language areas, Broca's and Wernicke's, which are usually found through fMRI, using T1w and RGB maps derived from DTI instead. The authors train two CNN-based architectures to perform the task using an in-house dataset of 30 epilepsy patients, and pretrain on the HCP dataset. The authors additionally introduce anatomical priors through the use of a custom loss function. The authors report Dice scores, ROC curves and perform a sensitivity analysis of their segmentations.

**Strengths:**

This paper is very well written and methodologically sound. The analysis of the results is thorough and the conclusion is apt. The authors described in sufficient details previous work, their data acquisition and labelling process, their training procedure and their results. The proposed method could be of interest for clinicians lacking fMRI acquisitions.

**Weaknesses:**

The main weakness of the paper is the limited real-world applicability through low performance and lack of open-source implementation. The reported Dice scores are indeed fairly low, although justified through the sensitivity analysis, and limit the confidence that one could have in the predicted segmentions. Moreover, even if the results were more accurate, the paper does not mention if the implementation of the method is open-source, if pre-trained models are available or if the dataset used could be downloaded.

**Detailed Comments:**

Some more minor comments:

- There is a typo in the title "with non-F**u**ctional Brain MRI"
- Page 3: "derived from DTI, encoding white **matter** tracts connectivity"
- The caption of Figure 1 should describe the content of the figure in more details.
- Page 4: "The Swin-UNETR was trained in a multiclass (two classes) setting with three output channels (background, Broca, Wernicke). AGYNet was trained in single-class mode with two output channels (background and Broca/Wernicke), separately for each class. Multiclass training of AGYNet was unstable and failed to converge." This section is confusing and could be reworded. It is unclear how or why the background has its own channel in the AHYNet formulation. What does the non-segmented part of either Broca or Wernicke's areas represent in their channel then ?
- Page 6: Pytorch, Adam and MONAI need citations
- Page 7: "Note that Dice was obtained with argmax applied to the network output channels." This is confusing wrt the output formulations on page 4.
-

**Justification Of The Final Rating:**

Even though the performance of the method is still dissapointing, the authors have done a great job of answering questions raised during my initial review, including the open-sourceness of the method and the inclusion of a new comparison method. My rating remains the same but it is now a stronger "4" than before. To push it to a 5, the authors would have had to either include better results or better justification for their low Dice.

Interestingly enough, in their rebuttal to my initial review, the authors mentionned this: "To the best of our knowledge, no prior work has tackled this ambitious task, nor are there established clinical datasets or benchmarks, positioning our study as a pioneering contribution. While there is room for improvement, our work establishes a strong foundation and serves as a benchmark for future research in this domain. Also, by nature, the task at hand deals with the segmentation of an “invisible” object, the functional activation, which is not directly observable neither in T1, nor in DTI but rather the result of cross-modality complex patterns."

This is a very strong argument in favour or their work and I wish, in hindsight, that a version of this paragraph had been included in the text, somewhere in the introduction and/or conclusion.

**Justification Of The Preliminary Rating:**

The paper is very well written, methologically sound and offers a thorough analysis of the results. The authors have described in great details their data acquisition and training procedures, and rigourously analysed their results, justifying their low performance.

**Questions To Address In The Rebuttal:**

Please address both major (open-sourceness and low performance) and minor (see above) comments.

**Special Issue:**

No

---

> ### Author Response · Authors · 2025-03-06
>
> Minor comments:
>
> All minor comment revisions in the manuscript have been highlighted in pink for convenience.
>
> Q1: There is a typo in the title "with non-Fuctional Brain MRI"
>
> A1: Thank you for your comment! We have addressed the issue accordingly.
> Please see the revised manuscript on page 1.
>
>
> Q2: Page 3: "derived from DTI, encoding white matter tracts connectivity"
>
> A2: Thank you for your comment! We have addressed the issue accordingly.
> Please see the revised manuscript on page 3.
>
>
> Q3: The caption of Figure 1 should describe the content of the figure in more details.
>
> A3: Thank you for your comment! We have addressed the issue accordingly. More details are thoroughly discussed in Sections 2.1, 2.1.1, and 2.3. We have revised the caption and included references to the relevant sections. Please see the revised manuscript on page 3.
>
>
> Q4: Page 4: "The Swin-UNETR was trained in a multiclass (two classes) setting with three output channels (background, Broca, Wernicke). AGYNet was trained in single-class mode with two output channels (background and Broca/Wernicke), separately for each class. Multiclass training of AGYNet was unstable and failed to converge." This section is confusing and could be reworded. It is unclear how or why the background has its own channel in the AHYNet formulation. What does the non-segmented part of either Broca or Wernicke's areas represent in their channel then ?
>
> A4: The reviewer is right in this comment. The inclusion of the extra non-segmented channel aligns with the original AGYnet architecture, as presented in the original paper. This design is a standard approach and is now more clearly explained in the revised manuscript. We have reworded this section for clarity. Please refer to the revised manuscript on page 4.
>
>
> Q5: Page 6: Pytorch, Adam and MONAI need citations
>
> A5: Thank you for your comment! We have addressed the issue accordingly.
> Please see the revised manuscript on page 6.
>
>
> Q6: Page 7: "Note that Dice was obtained with argmax applied to the network output channels." This is confusing wrt the output formulations on page 4.
>
> A6: The reviewer is right in this comment. The revision on page 4, as discussed earlier, may address this issue. The quoted sentence on page 7 has been removed.
>
>
>
>
> Answers to Questions To Address In The Rebuttal:
>
> All modifications and refinements are highlighted in blue.
>
> 1. A New Experimental Comparison with Existing Methods -
> We have added a new experiment to evaluate the performance of our algorithm against existing methods -
> This experiment assesses our approach in comparison to an atlas-based segmentation method using an fMRI-based atlas. We performed a 3D affine registration to align the atlas with each subject's coordinate system and measured the Dice coefficient against the ground truth. The results demonstrate that our approach significantly outperforms atlas-based segmentation, achieving a threefold improvement in the Dice coefficient. These findings highlight the limitations of current methods and underscore the advantages of our approach in achieving more accurate and subject-specific segmentation. See details in the revised manuscript, Sections 2.5, 3.3, and 4, highlighted in blue for convenience.
> “The comparison with the atlas-based method reveals the clear superiority of the proposed approach which provides subject-specific predictions.”
>
> 2. Performance Issues -
> Performance challenges arise from the indirect nature of our problem, data limitations, and clinical setting data. Comparison with existing non-fMRI methods highlights our significant improvements in the field. To the best of our knowledge, no prior work has tackled this ambitious task, nor are there established clinical datasets or benchmarks, positioning our study as a pioneering contribution. While there is room for improvement, our work establishes a strong foundation and serves as a benchmark for future research in this domain.
> Also, by nature, the task at hand deals with the segmentation of an “invisible” object, the functional activation, which is not directly observable neither in T1, nor in DTI but rather the result of cross-modality complex patterns.
>
> 3. Open-Sourceness -
> In accordance with the reviewer's request, we have created a GitHub repository containing the relevant code, implementation details, trained and pre-trained weights, atlas files, and more. A URL link is provided in the abstract section, and the repository will be made public upon acceptance.
> Unfortunately, due to regulatory issues, we are unable to provide public access to the dataset at this time.

---

> > ### Comment · Reviewer_Xk6b · 2025-03-14
> >
> > Thank you for the detailed response. My initial comments were addressed thoroughly, and the addition of the new experiment/comparison with the atlas is commendable. I have no further comments.

---

### Official Review · Reviewer_NGWa · 2025-02-19

**Confidence:** 4
**Preliminary Rating:** 4
**Recommendation:** Poster
**Final Rating:** 4

**Summary:**

This paper proposes a deep learning framework to map Broca and Wernicke, functional language areas of the brain, using only structural T1-w MRI and DTI data, eliminating the need for functional MRI for surgical planning. The authors implement two architectures—AGYnet and Swin-UNETR—and introduce an anatomically guided loss function to enforce all predictions to fall inside the gray matter. Only 30 fMRI epilepsy cases were used for testing. Results show high specificity but relatively low Dice scores, with improvements in sensitivity when applying safety margins (up to 5mm) for neurosurgical planning.

**Strengths:**

There exists clinical impact in that it reduces the dependence on fMRI brain mapping for pre-surgical planning. The authors combined both T1-w MRI and DTI as structural input, which provides detailed anatomical and spatial connectivity information.

A new loss function is proposed to penalize predictions of Broca and Wernicke area outside of the gray matter area.

The evaluation included cross-validation, ROC analysis, and Dice coefficient computation. Safety margins were also considered. Given 5mm safety margins, Broca had a sensitivity of 95%, and Wernicke had a sensitivity of 80%.

**Weaknesses:**

1. Even though the training for segmentation maps for Brodmann areas consists of 694 cases with T1w and DEC scans, they only had 30 fMRI epilepsy cases, which is very small.
2. The Dice coefficients (0.37 for Broca, 0.35 for Wernicke) indicate limited overlap between predicted and ground-truth maps. The sensitivity is also relatively low for the Wernicke area, even with a safety margin of 3mm-5mm.
3. The authors analyzed the impact of the anatomically-guided loss function, which only showed marginal increase in sensitivity and Dice coefficient for Broca area and showed no improvement of DICE coefficient and even lower sensitivity for Wernicke area.
3. The study does not compare against atlas-based approaches or previous non-fMRI deep learning models.

**Detailed Comments:**

The paper is well-written. See weakness for detailed comments.

**Justification Of The Final Rating:**

The authors have addressed all the questions raised in the initial review. However, the dataset's size is still limited by its clinical nature. Therefore, it raises concerns about the generalizability of this method.

**Justification Of The Preliminary Rating:**

The paper presents a clinically valuable and novel approach to mapping brain functional language areas without fMRI, with rigorous methodology and strong clinical motivation. However, concerns about small dataset size, segmentation and classification accuracy, and lack of comparison to alternative methods should be addressed.

**Questions To Address In The Rebuttal:**

1. It seems like Broca region oeverall have better response to the introduced anatomical loss term but Wernicke area did not improve with the loss function. Could you give an explaniation of why is that?

2. The paper only provided AUC for segmentation and classification tasks, however, no accuracy were shown. The ROC curve could be biased towards classes, leading to low accuracy but low AUC.

---

> ### Author Response · Authors · 2025-03-06
>
> All modifications and refinements are highlighted in blue for clarity.
>
> Weaknesses:
>
> Q1: Even though the training for segmentation maps for Brodmann areas consists of 694 cases with T1w and DEC scans, they only had 30 fMRI epilepsy cases, which is very small.
>
> A1: The dataset size is inherently limited by its clinical nature and complexity. The collection of triplet modalities—fMRI, T1w, and DWI—alongside expert-driven preprocessing and precise labeling. To the best of our knowledge, no comparable public dataset exists. Nevertheless, we conducted 5-fold cross-validation along with a detailed quantitative and qualitative analysis, which supports our claims and justifies our methodology.
> Q2: The study does not compare against atlas-based approaches or previous non-fMRI deep learning models.
>
> A2: Thank you for this comment! In agreement with the reviewer, we conducted a new experiment comparing our algorithm to an atlas-based segmentation method using a fMRI-based atlas. After performing 3D affine registration to align the atlas with each subject's coordinate system, we measured the Dice coefficient against the ground truth. Our approach achieved a threefold improvement in the Dice coefficient, significantly outperforming atlas-based segmentation. See details in the revised manuscript, Sections 2.5, 3.3, and 4.
>
> Q3: The Dice coefficients (0.37 for Broca, 0.35 for Wernicke) indicate limited overlap between predicted and ground-truth maps. The sensitivity is also relatively low for the Wernicke area, even with a safety margin of 3mm-5mm.
>
> A3: The results of the new experiment comparing our approach to atlas-based segmentation highlight the limitations of current methods and the advantages of our approach in achieving more accurate and subject-specific segmentation. Our method demonstrates a significant improvement in Dice coefficients compared to atlas-based approaches. Furthermore, sensitivity values of 80% and 95%, which align with acceptable margins for surgical procedures, underscore the clinical significance and practical applicability of our method. To the best of our knowledge, no prior work has tackled this ambitious, indirect task, nor are there established clinical datasets or benchmarks, positioning our study as a pioneering contribution.
> Also, by nature, the task at hand deals with the segmentation of an “invisible” object, the functional activation, which is not directly observable neither in T1w, nor in DTI but rather the result of cross-modality complex patterns.
> To support future research, we have created a GitHub repository containing the relevant code, implementation details, trained and pre-trained weights, atlas files, and more. The repository will be made public upon acceptance, serving as a valuable starting point for further advancements in this field.
>
> Q4: The authors analyzed the impact of the anatomically-guided loss function, which only showed marginal increase in sensitivity and Dice coefficient for Broca area and showed no improvement of DICE coefficient and even lower sensitivity for Wernicke area.
>
> A4: We performed an extended experiment to analyze the impact of the anatomically-guided loss function by repeating the evaluation on the AGYnet network, with the results. The updated findings provide new insights, showing an improvement in the Dice coefficient for both the Broca and Wernicke areas. See details in the revised manuscript on page 13.
> It is also important to note that the primary claim of our paper is the potential feasibility of localizing the Broca and Wernicke areas using non-fMRI input scans, while the anatomically-guided loss function serves as a secondary component.
>
> Questions To Address In The Rebuttal:
>
> Q1: It seems like Broca region overall have better response to the introduced anatomical loss term but Wernicke area did not improve with the loss function. Could you give an explanation of why is that?
>
> A1: The reviewer raises an interesting question, which is discussed in the revised manuscript on page 13.
> "The main intuition behind the L_Anatomy loss, as described in 2.4, is that differentiating between brain tissues may enhance segmentation accuracy… We believe that the complex anatomical structure of the Broca region, compared to the Wernicke region, makes it more responsive to anatomical guidance"
>
> Q2: The paper only provided AUC for segmentation and classification tasks, however, no accuracy were shown. The ROC curve could be biased towards classes, leading to low accuracy but low AUC.
>
> A2: The reviewer raises a valid point. To address this, we analyze the performance of our algorithm using multiple evaluation methods and metrics, including safety margin analysis, to provide a comprehensive and detailed assessment of our approach. This ensures a well-rounded evaluation beyond AUC, offering a complete picture of our model's performance.

---

> ### Comment · Area_Chair_Mtsq · 2025-03-14
> **Please review rebuttal and indicate final rating.**
>
> Dear Peiyu,
>
>     Since this paper has conflicting review scores and you have voted in favour of it, could you please go through the response and indicate whether your concerns have been addressed in the rebuttal?
>
> The official discussion period ends today, so please acknowledge the response even if you intend to keep your original rating.
>
> -Your MIDL AC

---

> > ### Comment · Reviewer_NGWa · 2025-03-14
> >
> > I have read the rebuttal, and I feel that my main concerns have been adequately addressed. Therefore, I am keeping my original score.

---

### Official Review · Reviewer_GemA · 2025-02-22

**Confidence:** 4
**Preliminary Rating:** 1

**Summary:**

The authors propose a novel approach to mapping Broca’s and Wernicke’s regions as a multimodal segmentation task, eliminating the reliance on fMRI. This alternative addresses the limitations of fMRI, including its time-intensive nature, the requirement for active patient participation, and the need for extensive manual post-processing and expert intervention.

**Strengths:**

The paper provides a thorough uncertainty quantification and safety margin analysis, offering valuable insights into the reliability of the proposed approach.

The authors present the brain mapping task in a clear and intuitive manner by formulating it as a multimodal segmentation problem. Additionally, the designed workflow is straightforward and easy to follow, enhancing the accessibility of the method.

**Weaknesses:**

The contribution of this work appears to be quite marginal. The primary novelty lies in the proposed combined loss function, which incorporates gray matter constraints alongside the standard Dice loss. However, there is no tailored design specifically addressing the segmentation of Broca’s and Wernicke’s regions. Ideally, any segmentation within the gray matter should incorporate additional constraints to penalize predictions extending beyond its boundaries. Moreover, the segmentation networks used in this study are pre-existing and widely adopted in segmentation tasks, limiting the methodological innovation.

Additionally, the absence of an ablation study on different input modalities leaves readers uncertain about the necessity of each modality. Based on Table 2, the introduction of the anatomy loss does not yield clear improvements—Wernicke’s sensitivity even decreases, while other metrics, such as specificity and the Dice coefficient, show no significant gains. The only notable improvement is a modest 2% increase in Broca’s sensitivity and Dice coefficient. Furthermore, the results in Table 2 are reported only for Swin-UNETR, leaving readers unclear on the impact of anatomy loss when using AGYnet. The lack of such comparisons adds to the ambiguity.

The work also lacks comparisons with related studies. Since this is essentially a multimodal segmentation problem, there should be existing multimodal segmentation methods available for benchmarking, even if their objectives are not perfectly aligned. Given the incremental nature of the proposed method, it is unlikely that no comparable work exists for evaluation.

Lastly, the dataset used is relatively small, and the paper does not clearly specify how it was split into training, validation, and testing sets. This raises concerns about the generalizability of the findings. Additionally, it is unclear why the HCP dataset was not utilized, and the potential limitations of excluding it should be addressed.

**Detailed Comments:**

see weakness

**Justification Of The Preliminary Rating:**

Given the current state of the work—lacking an ablation study and comparative benchmarking with related methods—along with its limited contribution (primarily an additional loss function term to the conventional Dice loss), the study does not demonstrate substantial novelty or impact. Furthermore, the reported improvements are marginal and evaluated on a very small dataset, raising concerns about the robustness and generalizability of the findings.

As it stands, the work is not yet suitable for publication in a venue like MIDL. I recommend the authors conduct further investigations, including more extensive experiments and comparative analyses, to strengthen the study’s contributions before resubmission.

**Questions To Address In The Rebuttal:**

see weakness

**Special Issue:**

No

---

> ### Author Response · Authors · 2025-03-06
> **We thank the reviewer for the comment and the effort invested in helping to refine our work. As requested, we conducted further investigations to strengthen the study’s contributions**
>
> We thank the reviewer for the comment and the effort invested in helping to refine our work. As requested, we conducted further investigations, including more extensive experiments and comparative analyses, to strengthen the study’s contributions before resubmission. We have submitted a revised manuscript with highlighted text for convenience.
>
> Q1: The work lacks comparisons with related studies. Since this is essentially a multimodal segmentation problem, there should be existing multimodal segmentation methods available for benchmarking.
>
> A1: In agreement with the reviewer, we conducted a new experiment comparing our algorithm to an atlas-based segmentation method using an fMRI-based atlas. After performing 3D affine registration to align the atlas with each subject's coordinate system, we measured the Dice coefficient against the ground truth. Our approach achieved a threefold improvement in the Dice coefficient, significantly outperforming atlas-based segmentation. See details in the revised manuscript, Sections 2.5, 3.3, and 4, highlighted in blue for convenience. The results of this experiment further highlight the limitations of current methods and the advantages of our approach in achieving more accurate and subject-specific segmentation, reinforcing its effectiveness.
>
> Q2: The proposed combined loss function, which incorporates gray matter constraints alongside the standard Dice loss. However, there is no tailored design specifically addressing the segmentation of Broca’s and Wernicke’s regions. Based on Table 2, the introduction of the anatomy loss does not yield clear improvements-Wernicke’s sensitivity even decreases. Furthermore, the results in Table 2 are reported only for Swin-UNETR, leaving readers unclear on the impact of anatomy loss when using AGYnet.
>
> A2: We performed an extended experiment to analyze the impact of the anatomically-guided loss function by repeating the evaluation on the AGYnet network. The updated findings provide new insights, showing an improvement in the Dice coefficient for both the Broca and Wernicke areas. We have also added intuition and discussion on the loss function’s objectives and provide an explanation for the differing behaviors between the Broca and Wernicke regions. See details in the revised manuscript on page 13
>
> Q3: The dataset used is relatively small, and the paper does not clearly specify how it was split into training, validation, and testing sets. This raises concerns about the generalizability of the findings
>
> A3:The dataset size is inherently limited by its clinical nature and complexity. The collection of triplet modalities—fMRI, T1w, and DWI—alongside expert-driven preprocessing and precise labeling of Broca and Wernicke areas is highly challenging and unique. To the best of our knowledge, no comparable public dataset exists, particularly in a clinical setting. Nevertheless, we conducted 5-fold cross-validation along with a detailed quantitative and qualitative analysis, which supports our claims and justifies our methodology. Regarding train-validation-test splitting, this information has been included in the revised manuscript on page 6. Additionally, we have provided a GitHub repository containing the relevant code, implementation details. The repository will be made public upon acceptance
>
> Q4: The absence of an ablation study on different input modalities leaves readers uncertain about the necessity of each modality.
>
> A4: The reviewer raises a valid point. Improved rephrasing has been included on pages 2 and 3 of the revised manuscript.
>
> Q5:It is unclear why the HCP dataset was not utilized, and the potential limitations of excluding it should be addressed.
>
> A5: Thank you for this comment. The HCP dataset was indeed utilized for our work. The phrasing on page 4, Section 2.3, has been improved and highlighted in blue, as the original text was somewhat unclear. Additionally, it is important to note that the HCP dataset does not provide explicit labels for the functional Broca and Wernicke areas.
>
> Q6: The primary novelty lies in the proposed combined loss function. Moreover, the segmentation networks used in this study are pre-existing and widely adopted in segmentation tasks, limiting the methodological innovation.
>
> A6: The primary claim and novelty of our paper lie in the potential feasibility of localizing the Broca and Wernicke areas using non-fMRI input scans, while the anatomically-guided loss function serves as a secondary component. To the best of our knowledge, no prior work has attempted this ambitious, indirect task, nor are there established clinical datasets or benchmarks, positioning our study as a pioneering contribution. While there is room for improvement, our work establishes a strong foundation and provides a benchmark, in clinical settings, for future research in this domain. To further support future research, we have created a GitHub repository serving as a valuable starting point for further advancements in this field

---

> ### Comment · Area_Chair_Mtsq · 2025-03-14
> **Please review response and submit final rating**
>
> Dear Haocheng,
>
>       Given that your initial recommendation on this paper was a strong reject and that other reviewers have voted in favour of this paper (albeit borderline), could you please indicate whether your initial concerns have been adequately addressed in the author rebuttal. The official discussion period ends today, so please acknowledge the response they have given even if you intend to stick to your rating.
>
> -Your MIDL AC

---

### Author Rebuttal · Authors · 2025-03-06

**Rebuttal:**

We have revised the manuscript and re-submitted it.

**Supporting Material:**

/attachment/0afd244351f69c54e4e5844d5b57e90dcd012200.zip

---

### Meta-Review · Area_Chair_Mtsq · 2025-03-20

**Recommendation:** Accept (Poster)
**Confidence:** 4

**Metareview:**

This paper had conflicting reviews with two reviewers voting in favour of acceptance and one rating it as a strong reject (said reviewer did not participate in the discussion phase).

Upon reading the reviews, author responses and the paper itself, many of the initial concerns raised seem to have been adequately addressed. While the segmentation model in itself does not offer significant novelty, I think that the task of isolating inherently `functional areas` such as language pathways in the brain makes for a challenging and interesting but seldom addressed problem setup, as opposed to purely anatomical areas. In this sense, the approach of pre-training with associated anatomical landmark targets (using a larger HCP dataset) and fine-tuning on the target epilepsy dataset seems like a reasonable starting point and the paper offers valuable insight into exploring this problem.

 While the limited target dataset size does raise questions about the robustness of the findings, I think the MIDL community would benefit from a presentation and discussion of this work in the conference forum, so that other researchers can build on its findings. In this sense, I vote in favour of including this paper in the conference

I strongly encourage the authors to include the following point which came up during the discussion in the actual paper "To the best of our knowledge, no prior work has tackled this ambitious task, nor are there established clinical datasets or benchmarks, positioning our study as a pioneering contribution. While there is room for improvement, our work establishes a strong foundation and serves as a benchmark for future research in this domain. Also, by nature, the task at hand deals with the segmentation of an “invisible” object, the functional activation, which is not directly observable neither in T1, nor in DTI but rather the result of cross-modality complex patterns.". This explanation serves as a valuable insight into why the problem is inherently challenging for a generic segmentation model to tackle. Additionally, by way of metrics, I would request the authors to perform significance testing on their AUCs since the performance of comparative models is rather close. Given the high AUCs couple with low dice, I think it makes sense to include additional metrics such as Hausdorff distance to qualify the findings from the paper.

I also hope that the authors hold up on their promise to release the code and data for the community to build onto their initial efforts in this direction.